# Design, Synthesis and Anti-Melanoma Activity of Novel Annexin V Derivative with β_3_-Integrin Affinity

**DOI:** 10.3390/ijms241311107

**Published:** 2023-07-05

**Authors:** Jingyi Zhu, Wenjuan Li, Jian Jing

**Affiliations:** Beijing Key Lab of Biotechnology and Genetic Engineering, College of Life Sciences, Beijing Normal University, Beijing 100875, China; 18630935132@163.com (J.Z.); 15132183225@163.com (W.L.)

**Keywords:** melanoma, Annexin V, β_3_-integrin, phosphatidylserine

## Abstract

Tumor tissues often exhibit unique integrin receptor presentation during development, such as high exposures of α_v_β_3_ and α_IIb_β_3_ integrins. These features are not present in normal tissues. The induction of selective thrombosis and infarction in the tumor-feeding vessels, as well as specific antagonism of α_v_β_3_ integrin on the surface of tumor endothelial cells, is a potential novel antitumor strategy. The Echistatin–Annexin V (EAV) fusion protein is a novel Annexin V (ANV) derivative that possesses a high degree of α_v_β_3_ and α_IIb_β_3_ integrin receptor recognition and binding characteristics while retaining the specific binding ability of the natural ANV molecule for phosphatidylserine (PS). We systematically investigated the biological effects of this novel molecule with superimposed functions on mouse melanoma. We found that EAV inhibited the viability and migration of B16F10 murine melanoma cells in a dose-dependent manner, exhibited good tumor suppressive effects in a xenograft mouse melanoma model, strongly induced tumor tissue necrosis in mice, and targeted the inhibition of angiogenesis in mouse melanoma tumor tissue. EAV exhibited stronger biological effects than natural ANV molecules in inhibiting melanoma in mice. The unique biological effects of EAV are based on its high β_3_-type integrin receptor-specific recognition and binding ability, as well as its highly selective binding to PS molecules. Based on these findings, we propose that EAV-mediated tumor suppression is a novel and promising antitumor strategy that targets both PS- and integrin β_3_-positive tumor neovascularization and the tumor cells themselves, thus providing a possible mechanism for the treatment of melanoma.

## 1. Introduction

Platelets are nucleated cell fragments produced by megakaryocytes, and their main function is to participate in coagulation and hemostasis. Increasing evidence has shown that platelets affect all aspects of tumor progression, especially metastasis of tumor cells [1]. Platelets contain many bioactive molecules that promote inflammation, cancer invasion, and metastasis. Tumor cells engage platelets in a process termed tumor cell-induced platelet aggregation, which aids tumor cells in evading the immune system. An increased number of platelets can promote cancer dissemination and immune evasion, and, intrinsically, more aggressive cancers can induce systemic inflammation and thrombopoiesis [2].

Platelets play an important role in tumor angiogenesis. When tumor cells detach from in situ tumor tissue and enter the vasculature, platelets are the first host component that contact tumor cells. Direct interactions of platelets and tumor cells result from the adhesive properties of tumor cells. One of the most important adhesive molecules involved in tumor cell-platelet interactions is platelet GPIIb-IIIa (fibrinogen receptor, integrin α_IIb_β_3_). Tumor-affected platelets isolated from mice with melanoma could migrate faster and longer and rapidly form thrombi [3]. Moreover, integrin β_3_ has been shown to play a major role in the adhesion process of B16F10 cells to platelets [4].

Integrin α_IIb_β_3_ is the main integrin expressed by platelets, along with integrins α_V_β_3_, α_2_β_1_, α_5_β_1_, and α_6_β_1_ [5]. Integrin α_IIb_β_3_-mediated cell adhesion is the main cause of platelet aggregation for thrombus formation. Because integrins play a key role in platelet aggregation, the inhibition of integrin α_IIb_β_3_ has emerged as an attractive strategy for the treatment of ischemic cardiovascular events. Currently, integrin α_V_β_3_ has also been increasingly studied. Integrin α_V_β_3_ is expressed at low levels in most normal cells but is highly expressed in various human tumor cell lines [6]. In addition, α_V_β_3_ is overexpressed on the surface of human malignant melanoma cells along with platelets [7]. Moreover, α_V_β_3_ integrin plays a crucial role in the progression of cutaneous melanomas from the benign radial growth phase to the metastatic vertical growth phase. Several earlier studies have shown that integrin α_V_β_3_ expression is upregulated or re-expressed during angiogenesis and that pathological angiogenesis is also associated with the upregulation of integrin α_V_β_3_ expression [8,9]. A mutant β_3_ integrin (Tyr747 and Tyr759) unable to undergo tyrosine phosphorylation was generated and neovascularization in vivo is impaired in mutant β_3_ knock-in mice [10]. This study showed that β_3_ integrin also forms a complex with VEGFR-2. The interaction of vascular endothelial cells with the melanoma-derived extracellular matrix (ECM) induces activation of the α_V_β_3_/VEGFR-2 signaling pathway, thus promoting angiogenesis; additionally, the inhibition of integrin α_V_β_3_ binding to the corresponding ligand effectively suppresses activation of the FAK-Src complex and VEGFR-2 [11]. 

Phosphatidylserine (PS) is externalized on tumor endothelial cells but not on endothelial cells in normal tissues [12,13,14]. In the tumor microenvironment (TME), exposed PS can also be found on tumor cells, secreted microvesicles, and tumor endothelial cells [14]. PS-mediated efferocytosis initiates a highly conserved process that prevents local and systemic immune activation by signaling by PS receptors. Importantly, PS receptor activation on immune cells creates an immunosuppressive milieu that tumor cells use as immune camouflage [15].

Annexin V (ANV) exhibits a high affinity for PS in the presence of Ca^2+^ and binds rapidly. This molecule has the highest affinity for PS among the many phospholipid-binding proteins, with a *K*_d_ of 0.1 to 2 nM [16]. PS-targeting monoclonal antibodies (mAbs) and ANV specifically localized to tumor vasculature but were not present in normal organs [17]. These observations suggest that targeting anionic lipids, such as PS, is feasible and has potential anticancer effects. PS-targeting mAbs initially developed to selectively bind to the tumor vascular system were found to have antitumor efficacy in tumor models. Yin et al. found that PS-targeting mAbs dramatically shifted the phenotype of macrophages from an M2-like to an M1-like phenotype and that the mAb induced the differentiation of myeloid-derived suppressor cells (MDSCs) to M1-like macrophages and mature dendritic cells (DCs_ and reduced the expansion of immunosuppressive cell types, including MDSCs and Tregs, in the TME) [18].

PS-targeting mAbs could facilitate an adaptive immune response [19]. The combination of PS-targeting mAbs with immune checkpoint blockade (anti-cytotoxic T-lymphocyte associated protein 4 (CTLA-4) or anti-PD-1) was evaluated in breast and melanoma syngeneic models of cancer in immunocompetent mice [20,21]. PS targeting enhanced the efficacy of anti-PD-1 and altered the immune landscape of tumors by increasing T-cell infiltration, proliferation, and activation [21]. These results suggest that the anticancer efficacy of PS-targeting mAbs results from targeting the tumor vasculature and altering the immune microenvironment of tumors by interfering with PS-mediated immune suppression. PS targeting modifies PS-mediated immunosuppression and promotes the induction of innate and adaptive antitumor immune responses. Echistatin (Ech) is an RGD-type disintegrin isolated from the Viperidae venomous snake *Echis carinatus* with a high affinity for integrins α_V_β_3_ and α_IIb_β_3_. In in vitro experiments, Ech inhibited the proliferation, migration, invasion, and adhesion of highly metastatic human osteosarcoma cells overexpressing α_V_β_3_ integrin [22]. We previously constructed a fusion protein containing the structural domains of Ech and ANV (EAV), and preliminary studies showed that this protein exhibited better antithrombotic ability [23]. Our subsequent studies showed that the fusion protein also exhibited better a_V_β_3_ integrin receptor recognition activity. Herein, we more systematically present the characteristics of the fusion protein exhibiting inhibition of melanoma in mice.

## 2. Results

### 2.1. Preparation and Simulation of Recombinant EAV Fusion Protein

We predicted the three-dimensional structure of EAV by using the AlphaFold online website [24], and the prediction results initially showed that the fusion protein could better simulate the Ech active structural site while preserving all of the structural regions of ANV (Figure 1a,b). Further structural determination is in progress, and although the simulation results were not fully representative of the true structure of the fusion protein, they indicated that the novel fusion protein formed by fusing Ech and ANV does have the potential to exhibit the biological activities of both proteins. 

We constructed an expression system containing EAV by using gene recombination technology, after which we prepared and purified the fusion protein. GST-EAV cleaved by PreScission Protease (PSP) was subjected to affinity chromatography to obtain the EAV protein. The EAV and ANV proteins were detected by using 12% SDS-PAGE electrophoresis (Figure 1c,d), and the EAV protein was a monomeric protein with a purity >95%. We examined the purified EAV fusion protein via Western blotting with a mouse polyclonal antibody against the natural ANV protein. The results showed that EAV was recognized and bound by the antibody (similar to the natural ANV molecule) compared to the natural ANV, indicating that EAV has the same antigenic determinant cluster as ANV.

The secondary structures of ANV and EAV were identified by circular dichroism (CD) (Table 1). As reported in previous articles, the primary secondary structure of ANV is comprised of α-helices. CD showed that the secondary structure species of EAV did not change, and each secondary structure increased in number, thus indicating that the secondary structure of ANV was well preserved. The amino acid residue number of the α-helix in EAV did not decrease but increased by six. Moreover, the RGD motif, which mediates Ech functions, was located within a β-turn [25]; in addition, compared to that of ANV, the β-turns of EAV increased by eight. This finding suggested that the secondary structure of ANV and Ech in EAV was more complete, which essentially confirmed the dual function of ANV and Ech. 

### 2.2. Integrin Specificity and Phospholipid Binding Properties

Furthermore, we examined the EAV fusion protein for cell adhesion. In previous studies, we found that the fusion protein EAV shows better recognition of and binding to platelet-exposed α_IIb_β_3_ integrin receptors. In this study, we selected HT29 cells with high expression of integrin α_V_β_3_ and PC12 cells highly expressing integrin α_1_β_1_ to assay the biological characteristics of EAV adhesion to the two different cell types. The results of the experiments are shown in Figure 2a–c. PC12 cells incubated on wells coated with CoI-IV showed a significant increase in absorbance values. This result indicated that PC12 cells were bound to the wells coated with CoI-IV. When PC12 cells coincubated with EAV and ANV were added to wells coated with CoI-IV, the absorbance values did not significantly change, thus indicating that EAV and ANV did not prevent PC12 from binding to CoI-IV and laterally verifying that the integrin site on PC12 responsible for binding to CoI-IV was not masked. Subsequently, we constructed a recombinant expression system for Ech, and GST-Ech protein was purified to high homogeneity by using affinity chromatography (Figure 2d). The findings in Figure 2e show that the number of HT29 cells increased in the wells coated with fibronectin (FN). Moreover, the number of HT29 cells incubated with EAV or Ech decreased with increasing protein concentration (Figure 2e), and the absorbance values of cell lysates also reflected this phenomenon (Figure 2b), thus demonstrating that the binding site of HT29 to FN was covered by EAV or Ech. The number of HT29 cells was essentially not reduced after ANV incubation (Figure 2b). In addition, when the concentrations of EAV and Ech were both 10 nM, the absorbance values were both substantially different from those of the ANV group (Figure 2c). Furthermore, combined with the results of previous studies, the data showed that EAV better preserved the recognition and binding ability of α_IIb_β_3_ receptors on platelets [23]. These characteristics were very similar to the integrin receptor binding characteristics of Ech, thus suggesting that EAV retains the main functional activity of Ech.

In previous studies, we examined the phospholipid binding ability of EAV [23]. EAV molecules exhibited good recognition and binding ability of PS but did not bind phosphatidylcholine (PC) [23]. This feature was similar to that of the natural ANV molecule. The results indicated that the ANV functional structural domain in the EAV molecule was well preserved and could effectively demonstrate the corresponding PS-specific recognition and binding characteristics. 

The in vitro molecular binding characteristics of the EAV fusion protein also imply that both its Ech functional motif and ANV functional region were effectively conserved to fully display the functional conformation and to exhibit the corresponding biological functions.

### 2.3. EAV Inhibits the Proliferation and Migration of Melanoma B16F10 Cells

We examined the effects of EAV on the viability of B16F10 cells by using the MTT assay, and the results are shown in Figure 3a,b. EAV at a high concentration significantly reduced the viability of B16F10 cells, which tended to decrease in a dose-dependent manner. At the same concentration of ANV and EAV protein, EAV showed a better ability to inhibit B16F10 cell viability, both for 24 h and 48 h. The effects of EAV on the migration of B16F10 cells were investigated via a cell scratch assay, and the results of the study showed (Figure 3c,d) that at 24 h, EAV inhibited the migration of B16F10 cells in a dose-dependent manner.

We also assessed apoptosis by treating B16F10 cells with different concentrations of EAV and ANV for 24 h by using flow cytometry. The results showed that the apoptosis rates of the experimental and control groups were not significantly different and did not increase with increasing protein concentration, which indicated that EAV and ANV were not able to induce apoptosis of B16F10 cells in the abovementioned concentration range.

We examined whether EAV affects VEGF expression in B16F10 cells by using Western blotting. B16F10 cells cultured in vitro expressed VEGF-A. VEGF-A is a dimer with a monomeric molecular weight of 27 kDa. In addition, α-Tubulin (with a molecular weight of 50 kDa) was used as an internal reference. After treatment of B16F10 cells with different concentrations of ANV and EAV for 24 h, Western blotting was performed to assess the expression of VEGF-A in B16F10 cells. As shown in Figure 3e,f, there was no significant difference in the expression of VEGF-A in the control, ANV, and EAV groups, thus confirming that ANV and EAV did not significantly affect the expression of VEGF-A in B16F10 cells in the concentration range of 0.1–10 nM.

### 2.4. Inhibition of Tumor Growth Rate in a Mouse Xenograft Model by EAV

We performed Xenograft experiments to investigate whether ANV and EAV have anti-melanoma effects. A mouse melanoma model was first established and continuously treated for 13 days. The anticancer drug DITC was used as a positive control. The DTIC group showed the lowest weight gain. Moreover, during the first 11 days of treatment, the growth rate of the transplanted melanoma volume in the high-dose EAV group remained essentially the same as that in the DTIC group (Figure 4a). The growth rate of the tumor volume in the negative control group (PBS) was higher than that in all of the experimental groups, and the growth rate of the tumor volume in the positive control group (DTIC) was the slowest, whereas the growth rate of the tumor volume in the ANV and EAV groups was inhibited (to a certain extent) and was dose-dependent. Additionally, the inhibitory effect of EAV was higher than that of ANV. As shown in Figure 4b, with the increase in tumor volume, the body weight of the mice in the PBS group steadily increased, and the body weight of the mice in the DTIC group increased most slowly, which indicated that this treatment was more toxic to the mice. ANV and EAV were able to maintain the body weight of mice better than DTIC, thus indicating that ANV and EAV were less toxic. The body weight of the mice in the ANV group increased more than that in the EAV group, thus indicating that EAV had a stronger inhibitory effect on tumor growth and had a more obvious effect on the body weight of mice compared with those in the ANV group.

### 2.5. EAV Induced Melanoma Tumor Necrosis In Vivo

The H&E staining results demonstrated dense tumor tissue in the PBS group (Figure 4c). The tumor tissue in the DTIC group had a large necrotic area and loosely arranged cells. Additionally, the DTIC group exhibited poorly defined cells with red-stained cytoplasm and mostly fragmented nuclei. Similarly, significant areas of necrosis were observed in the experimental group, with more tumor necrosis observed in the EAV group than in the ANV group. The EAV and ANV groups exhibited tumor necrosis in a dose-dependent manner. Moreover, the tumor cell spacing in the necrotic area was larger and most obvious in the EAV high-dose group. The tumor cells in the necrotic area were widely spaced and accompanied by some bleeding, with increased neutrophil infiltration. In addition, the nuclei of the tumor cells in the necrotic area were solidified, broken, or even lysed, and the cell fragments were mostly red in color. The experimental results showed that EAV increased the necrosis of mouse melanoma tissues, and the necrosis of B16F10 tumor tissues was stronger than that of the ANV group.

### 2.6. EAV Specifically Inhibited Angiogenesis in Mouse Melanoma Tumors

Angiogenesis is a crucial part of the tumor growth process, and we investigated whether the mechanism of action of EAV in inhibiting tumor cell growth was related to the inhibition of angiogenesis. Desmin is an intermediate filament protein expressed in vascular smooth muscle, and tumor angiogenesis was observed by using a fluorescence localization assay of Desmin to assess the vascular density of B16F10 tumor sections. As shown in Figure 4d, the highest vascular density in tumor sections was observed in the PBS group, and a decrease in vascular density could be observed in the ANV low-concentration group; the density significantly decreased with increasing ANV concentration. Moreover, the vascular density was significantly decreased in the EAV group, and the inhibition of angiogenesis was stronger in the EAV low-concentration group than in the ANV low-concentration group in a dose-dependent manner. Therefore, we demonstrated that the inhibition of B16F10 tumor growth by EAV is related to its inhibition of angiogenesis.

### 2.7. Targeting of EAV to Melanoma Tumor Tissue in Mice

To verify whether EAV acts directly on tumor tissue, we injected tumor-bearing mice via tail vein injections using FITC fluorescent-labeled protein. As shown in Figure 5, the accumulation of ANV and EAV at the tumor location gradually increased with increasing concentration. Low concentrations of EAV were present in small amounts in tumor tissue, and low concentrations of ANV did not affect tumor tissue. In the medium- and high-concentration groups, the ANV distribution was scattered, and EAV demonstrated an aggregated distribution. Furthermore, the fluorescence brightness of the EAV group was stronger than that of the ANV group, which indicated that EAV may accumulate in higher amounts in the tumor tissue. Notably, it seems that 834 nmol/kg EAV is distributed along the route where the tumor tissue is destroyed.

## 3. Discussion

Because platelets within the normal physiological range can support tumor progression, the use of antiplatelet therapies has been posited as a strategy to interdict or prevent the signaling events that drive cancer growth and metastasis [26]. Antithrombotic therapies, such as aspirin, warfarin, and cyclooxygenase (Cox) inhibitors, can improve cancer patient survival in some cases [10]. In a mouse tumor model, antiplatelet agents were shown to reduce tumor metastasis. Platelets adhering to the surface of tumor cells not only facilitate the survival of tumor cells in the blood but also mediate the adhesion of tumor cells to vascular endothelial cells, thus promoting the migration and arrest of tumor cells in the blood vessels. EAV is a recombinant fusion protein with antithrombotic effects, and it consists of two parts: Ech (which is a member of RGD disintegrin) and ANV.

The arginine-glycine-aspartate (RGD) peptide is one of the most effective and widely used compounds for the delivery of imaging agents and drugs to tumors [27]. Surface-embedded Ech microbubbles target α_v_β_3_-integrins, and such microbubbles have been used to detect early tumor angiogenesis [28]. The RGD receptor integrin α_v_β_3_ plays a key role in the regulation of tumor growth and metastasis and tumor-mediated angiogenesis. The interaction of VECs with melanoma-derived ECM was shown to induce the activation of the α_v_β_3_/VEGFR-2 signaling pathway, thereby promoting angiogenesis, and the inhibition of integrin α_v_β_3_ binding to the corresponding ligands effectively represses the activation of the FAK-Src complex and VEGFR-2 [11]. In A375 cells, the aggressiveness of melanoma was reduced if the activity of β_3_ integrin was inhibited. Integrin α_v_β_3_ mediates c-Src phosphorylation and activates the PI3K/Akt signaling pathway by interacting with the ECM, thus participating in the tumor proliferative process. RGD-type disintegrin can interfere with the α_v_β_3_/VEGFR2 signaling pathway by binding to integrin α_v_β_3_ and inhibiting the interaction of α_v_β_3_ with ECM [29]. Ech, which is an RGD-type disintegrin, inhibited platelet aggregation and (to some extent) the adhesion of tumor cells to FN, LN, and vitronectin (VTN). Integrin receptors (especially α_v_β_3_) are overexpressed in tumor cells and involved in tumorigenesis, and related studies have exploited the disintegrin-specific binding of integrins to design drug molecules containing RGD motifs that exhibit the potential to target binding to relevant integrin sites in the tumor vasculature. Moreover, nanoparticles coupled to RGD peptides are highly targeted to α_v_β_3_ integrins, which is considered to have potential for angiogenic imaging [30] and for the delivery of targeted therapeutic oncology drugs in tumor tissue targeting α_v_β_3_ [31]. Compared to other disintegrins, Ech is a broader spectrum disintegrin that targets a variety of integrins, especially β_3_-type integrins such as α_v_β_3_ and α_IIb_β_3_. EAV effectively binds β_3_-type integrins through its Ech structural region and exerts effective biological effects to inhibit melanoma in mice.

Exogenous ANV has been shown to block the transmission of oncogene-containing microvesicles between tumor cells and between tumor cells and VECs in a mouse melanoma model, thus impeding tumor angiogenesis [32]. Previous studies have demonstrated that the overexpression or downregulation of endogenous ANV affects tumorigenesis; however, ANV plays different roles in different tumors [33]. ANV constitutes a fusion protein with related antitumor proteins as part of the development of ANV-based antitumor drugs. Based on its high affinity for PS, the antitumor protein is targeted to the tumor region to further enhance the antitumor activity. The combination of ANV with other immune checkpoint inhibitors could significantly enhance antitumor activity [34]. Furthermore, purine nucleoside phosphorylase (PNP) fusion protein with ANV enhanced cytotoxicity against breast cancer [35]. This finding suggests that ANV, as a PS recognition protein, may be used to deliver specific cytotoxic drugs that selectively destroy blood vessels in solid tumors, thus indicating that it is a novel inhibitor of angiogenesis in tumor therapy.

Currently, four genomic subgroups of melanoma have been identified through studies of The Cancer Genome Atlas (TCGA) network: BRAF (B-rapidly accelerated fibrosarcoma), NRAS, NF1, and triple wild type [36]. Mutation patterns, including the total number of mutations and the type of driver oncogene, vary by melanoma subtype. Most of the current targeted therapeutics for melanoma disease have been developed on this basis. For example, the combined use of BRAF and MEK inhibitors was applied in the treatment of patients with advanced BRAF-mutated melanoma. The BRAF inhibitors vemurafenib and dabrafenib were approved for the treatment of metastatic and unresectable BRAF-mutated melanoma in 2011 and 2013, respectively [37,38]. KIT inhibitors may also be a targeted therapy option. Although these agents were effective in approximately half of the patients with BRAF-mutated melanoma, most patients developed secondary resistance in a relatively short period [39]. The mechanisms of this secondary resistance are also being explored [40,41], and novel drugs or combinations of new drugs are being developed for the treatment of melanoma.

Herein, we investigated whether a fusion protein with an antiplatelet effect (EAV) has an anti-melanoma effect. EAV has two motifs: the RGD motif of Ech and the PS-binding motif of ANV. Based on the simulation of spatial structure, secondary structure determination data, and assay results from cell adhesion experiments, the functional structural regions of Ech were well presented and were able to exhibit integrin receptor binding characteristics that were almost identical to those of the natural Ech. EAV has the capacity to recognize and bind α_IIb_β_3_, in addition to effectively binding α_v_β_3_. α_IIb_β_3_ receptor antagonists are the most potent antiplatelet agents, and several agents have been approved for clinical practice, such as abciximab, tirofiban, and eptifibatide. Eptifibatide was shown to reduce the invasive ability of MDA-MB-231 human breast cancer cells and inhibit the adhesion of MDA-MB-231 cells to endothelial cells [42]. Eptifibatide and abciximab can also promote apoptosis in MCF-7 human breast cancer cells. As a potent α_IIb_β_3_ receptor antagonist, EAV also possesses the ability to inhibit tumor cell–platelet interactions by binding α_IIb_β_3_. However, our study also demonstrated that in terms of phospholipid binding characteristics, EAV fusion proteins could approach the natural state ANV. Therefore, it is reasonable to believe that the EAV fusion protein can also effectively interfere with the biological behavior of B16F10 cells by specifically acting on many exposed PS molecules on the surface of tumor cells to inhibit melanoma development. Our study confirmed that EAV was effective in slowing the growth of melanoma volume, enhancing the destruction of melanoma tumor tissue, and inhibiting the development of blood vessels in tumor tissue. Moreover, EAV significantly targeted mouse melanoma tumor tissues, effectively enhancing its antitumor biological effects. These results suggest that EAV has the potential to become a new therapeutic agent for melanoma. In future studies, we will also explore the relevant molecular signaling pathways to better understand their refined mechanisms of action. Additionally, we believe that pharmacokinetic studies are necessary.

## 4. Materials and Methods

### 4.1. Reagents, Chemicals, and Antibodies

Bovine serum albumin (BSA), MG132 proteasome inhibitor, cathepsin inhibitor-I, DMSO, and crystal violet were obtained from Millipore Sigma (St. Louis, MO, USA). The α-Tubulin rabbit polyclonal antibodies, Desmin rabbit monoclonal antibody, donkey anti-rabbit Alexa Fluor 555, and HRP-labeled IgG were purchased from Beyotime (Shanghai, China). VEGF rabbit polyclonal antibodies and ECL luminescence kits were purchased from Boster (Wuhan, China).

### 4.2. Protein Preparation and Purification

Recombinant EAV and ANV were expressed and purified as previously described [5]. To obtain the soluble recombinant protein in the *E. coli* system, the fusion protein was attached to the C-terminus of the GST fusion tag. PCR was used to amplify the Ech genes. The target gene was inserted into the pGEX-6P-1 vector between the BamHI and XhoI endonuclease sites. In addition, the gene fragment containing Ech was ligated to the vector and transferred to *E. coli* BL21 (DE3) for expression, after which GST-Ech was purified by using GST affinity chromatography. The quality of the protein was assessed by using 12% sodium dodecyl sulfate-polyacrylamide gel electrophoresis (SDS-PAGE) followed by Coomassie blue staining.

The primers that were used to amplify the target fragment of Ech were as follows:

ECHF: 5′-TTCCAGGGGCCCCTGGGA-3′

ECHR: 5′-ACGATGCGGCCGCTCGAGTTAGGTAGCCGGACC-3′

### 4.3. Circular Dichroism

CD was used to determine the secondary structure of the recombinant protein. The purified recombinant proteins were dialyzed in 20 mM PB buffer (pH 7.4) and then diluted to the same concentration. The CD measurement wavelength was between 190 and 260 nm. 

### 4.4. Cell Culture

The murine melanoma cell line B16F10 was purchased from DingGuo (Beijing, China). Cell lines were cultured in DMEM (M&C Gene Technology, Beijing, China) with 10% fetal bovine serum (FBS) and incubated in a humidified atmosphere at 37 °C in 5% CO_2_. HT29 and PC12 cells were cultured as previously described [43].

### 4.5. Cell Adhesion Assay

The effects of Ech, EAV, and ANV on HT29 and PC12 cell adhesion were determined. Briefly, 96-well plates were coated with 20 μg/mL FN and 10 μg/mL Col-IV and incubated for 12 h at 37 °C. The protein solution in the wells was then recovered, and 100 μL of 1% BSA solution was added to each well for 2 h. A total of 5 × 10^4^ cells were mixed with different proteins in 1.5 mL EP tubes with protein concentrations set at 0.01 nM, 0.1 nM, 1 nM, and 10 nM. After incubation for a certain period, the abovementioned solution was added to 96-well plates containing cells for incubation. PBS was washed to remove nonadherent cells, followed by fixation and staining. The number of adherent cells was determined by measuring the optical density at 600 nm via a PoLARstar Omega enzyme marker (BMG, Berlin, Germany).

### 4.6. Cell Proliferation Assay

Exponential growth phase B16F10 cells were collected, and 2000 cells per well were cultured in 96-well plates and incubated overnight. Cells were treated with ANV and EAV for 24 h and 48 h, respectively. Protein concentration gradients of 0.1 nM, 1 nM, and 10 nM were set. The in vitro cytotoxic effects of these treatments were determined via MTT assays (570 nm).

### 4.7. Wound Healing Assay

An in vitro scratch-wound healing assay was used to study cell migration. B16F10 cells were seeded in 6-well plates (3 × 10^5^ cells/well); and after adhesion, linear scratch wounds were made in the cell monolayers with a 200 μL pipette tip. The cells were then stimulated with ANV and EAV for 24 h (37 °C and 5% CO_2_), and the scratch wounds were visualized with an inverted microscope (Zeiss, Germany). The area of each scratch wound was determined using ImageJ 1.8.0 software.

### 4.8. Apoptosis Detection by Flow Cytometry

B16F10 cells (2 × 10^5^) were inoculated in 6-well plates at 2 mL per well and incubated overnight (37 °C, 5% CO_2_). The concentrations of ANV and EAV were 0.1 nM, 1 nM, and 10 nM. Protein solutions of different concentrations were added to six-well plates and coincubated with cells for 24 h. The cells were collected by trypsinization and centrifugation at 800 rpm for 5 min. Following resuspension in binding buffer, a single-cell suspension was incubated with 5 μL of Annexin-V–FITC (Dingguo, Beijing, China) and 5 μL of propidium iodide (PI) (Dingguo, Beijing, China) for 15 min at room temperature in the dark. Finally, apoptosis status was detected by NovoCyte3130 flow cytometry (ACEA, Arcadia, CA, USA).

### 4.9. Western Blotting

Trypsin digestion of 2 mL of B16F10 cells in logarithmic growth phase was inoculated in 6-well plates (cell density of 4 × 10^5^ cells/mL) and cultured overnight (37 °C, 5% CO_2_). Protein solution was added, and the experiment was set up for the blank and experimental groups (EAV and ANV final concentrations of 0.1 nM, 1 nM, 10 nM), and the culture was continued for 24 h. After the cells were collected, 100 μL of cell lysate was added to extract the total protein, and the protein concentration was determined via the BCA method. SDS-PAGE electrophoresis (5% concentrated gel, 10% separated gel) was performed, and PVDF membrane conditions were transferred (ice water bath, 180 mA, 100 min). The primary antibody was incubated at 4 °C for 14 h, and the secondary antibody was incubated at room temperature for 1 h. After incubation, the antibody was washed three times with TBST, and ECL luminescence was performed. An enhanced chemiluminescence system (Tanon, Shanghai, China) was used to detect protein expression.

### 4.10. Xenograft Animal Model

Seven-week-old female C57BL/6J mice (5 mice per experiment, 18–20 g) were obtained from Beijing Vital River Laboratory Animal Technology Co., Ltd. (Beijing, China). Mice were housed in a pathogen-free air barrier facility with a temperature range from 20 to 24 °C. All of the animals had continuous access to food and water with no restrictions, and all of the handling methods and procedures were approved by the Animal Care and Use Committee of the College of Life Sciences, Beijing Normal University. Tumor xenograft models were generated via subcutaneous injections of B16F10 cells (4 × 105 cells per mouse) into the right dorsum of the mice. Each group of mice was injected with PBS, DTIC, EAV (139 nmol/kg), EAV (278 nmol/kg), ANV (139 nmol/kg), or ANV (278 nmol/kg) for 13 days. The mouse activity was observed daily, and their body weight and tumor growth measurements were recorded. Tumor volume was determined as mm3 using the equation A × B2 × 0.52, where A is the length (mm) and B is the width (mm) [44].

### 4.11. H&E Staining

Dissected tumor tissues were preserved in 4% paraformaldehyde (Dingguo, Beijing, China) for fixation overnight at 4 °C. The fixed tumor tissue was cut to the appropriate size and placed in an embedding box immersed in different concentrations of ethanol for gradient dehydration. Afterward, the samples were transparently transferred to xylene, placed in melted paraffin wax (60 °C, 2 h), changed once overnight, and embedded and sectioned (5 μm thickness) the next day by using a paraffin embedding machine (Leica, Germany) and a paraffin sectioning machine (Leica, Germany). The sections were dewaxed, rehydrated and stained, sealed with a drop of neutral resin (Dingguo, Beijing, China) at the edge of the sections, and observed under an inverted microscope.

### 4.12. Immunohistochemical Examination (IHC)

Paraformaldehyde-fixed paraffin-embedded tissue samples were dewaxed by using a reduced xylene/alcohol series followed by thermally induced antigen recovery (pH 6). Subsequently, specimens were treated with peroxidase blocking solution (DingGuo, Beijing, China), blocked with 4% BSAT and incubated with primary antibodies against Desmin for 12 h. After incubation with the appropriate HRP-labeled secondary antibody for 2 h, the tablets were sealed with a DAPI-containing sealer under light-proof conditions. Pictures were taken with an inverted fluorescence microscope (Zeiss, Germany).

### 4.13. Fluorescence Assay Based on FITC-Labeled EAV

Immunofluorescence assays for the localization of fusion proteins in melanoma tissues were performed. FITC was added to crosslink the fusion protein (mProtein:mFITC = 100:1). The proteins were then packed into MD34 dialysis bags (MWCO3500) (TaKaRa, Japan) and dialyzed against PBS for 24 h (4 °C, 12 h followed by a buffer change). The FITC-labeled proteins were added to a dextran gel G25 (TaKaRa, Japan) column (1 mm × 400 mm) to separate the free FITC. Finally, FITC-labeled protein was injected into the tail vein of melanoma mice; 30 min after injection, tumor tissue sections were produced. The samples were observed via laser scanning confocal microscopy.

### 4.14. Statistical Analysis 

Statistical analysis of experimental results was performed by GraphPad Prism 9.0 (GraphPad Software, Inc., La Jolla, CA, USA). All statistics are shown as mean ± SD from at least three independent experiments. Unpaired Student’s *t*-test was used to assess differences between two groups, while analysis of variance (ANOVA) was used for comparisons of multiple groups. *p* < 0.05 was considered a significant difference. The level of significance is indicated by asterisks (* *p* < 0.05, ** *p* < 0.01, *** *p* < 0.001, and **** *p* < 0.0001). 

## 5. Conclusions

We concluded that a recombinant antithrombotic ANV derivative (EAV) with β_3_-type integrin affinity exhibits a greater ability to inhibit the progression of melanoma in mice than natural-state ANV. The multiple mechanisms provide compound effects that facilitate the antitumor effects of single molecules. This effect may suggest new strategies for future antitumor drug development, especially anti-melanoma drug development.

## 6. Patents

The research reported in this manuscript has been declared a China Invention Patent.

## Figures and Tables

**Figure 1 ijms-24-11107-f001:**
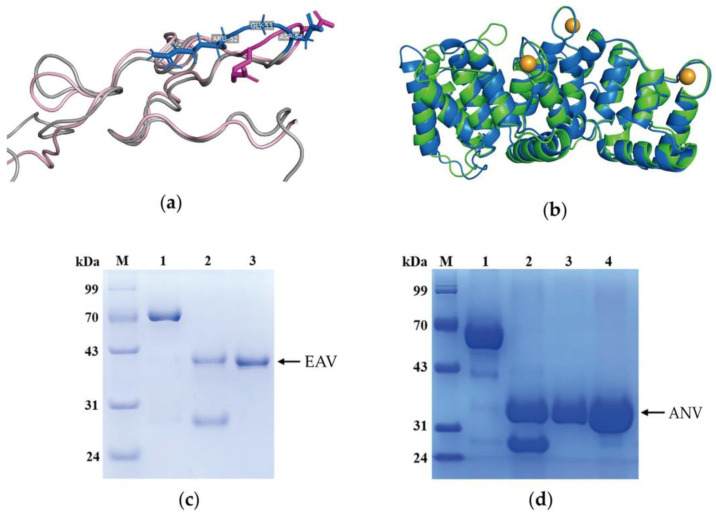
Structural prediction and preparation of the recombinant fusion protein EAV: (**a**) Comparison of the Ech structural domain of EAV (gray) with the natural Ech (pink) structure; the RGD motif of EAV is blue, and for Ech, the RGD motif is dark pink. (**b**) Comparison of the ANV structural domain of EAV (blue) with that of ANV (green); the orange spheres are calcium ions. (**c**) Purified EAV fusion protein detected via 12% SDS-PAGE; Lane 1 is GST-EAV (70.9 kDa); Lane 2 is EAV after PSP digestion (the molecular weight of EAV is 44.5 kDa, and the molecular weight of GST is 26 kDa); Lane 3 is EAV. (**d**) Purified ANV detected by using 12% SDS-PAGE; Lane 1 is GST-ANV (62.7 kDa); Lane 2 is ANV after PSP digestion; Lane 3 and Lane 4 are ANV (36.3 kDa).

**Figure 2 ijms-24-11107-f002:**
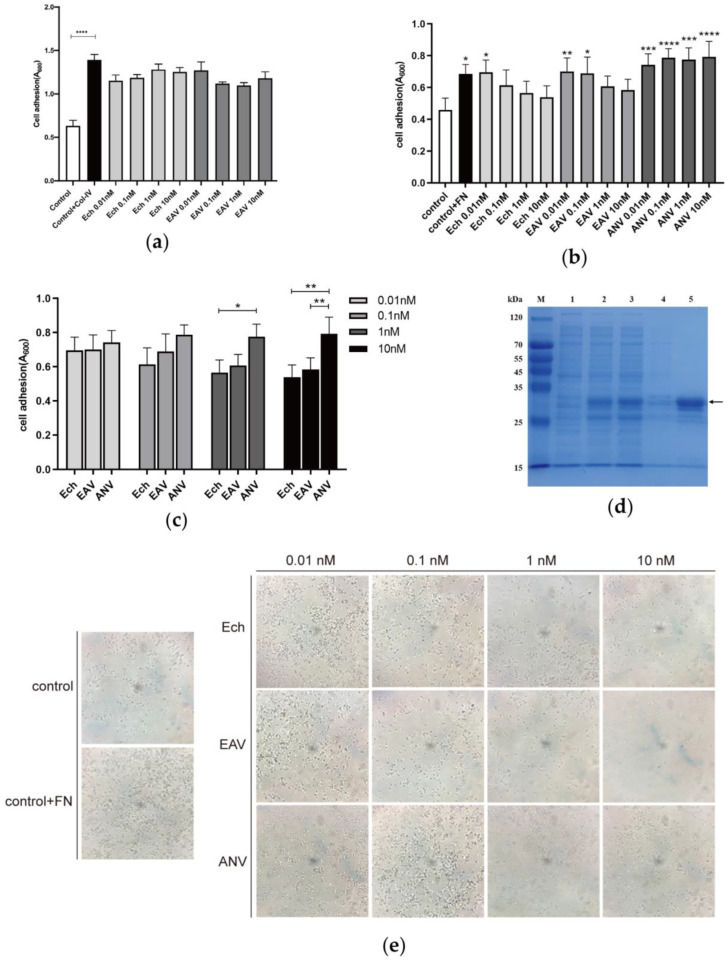
EAV cell adhesion analysis: (**a**) With PC12 cells as the experimental material, sample wells coated with Col-IV and uncoated wells as negative controls were used, and the concentration gradients were set to 0.01 nM, 0.1 nM, 1 nM, and 10 nM with reference to the IC50 values of Ech on other tumor cells [16]. Subsequently, the OD_600_ value was measured. (**b**) With HT29 cells as the experimental material, sample wells coated with FN and uncoated wells as negative controls were used. *p* value indicates the experimental group compared to the control group. (**c**) Comparison of HT29 cells treated with the same concentration of Ech, EAV, and ANV. (**d**) Preparation and purification of Ech. The pGEX-6P-1-Ech recombinant expression vector was constructed and transformed into *E. coli* BL21 (DE3) for expression. Lane 1 lacked IPTG-induced pGEX-6P-1-Ech expression of whole protein; Lanes 2–4 included IPTG-induced pGEX-6P-1-Ech expression of whole protein, supernatant, and precipitate; Lane 5 included GST-Ech (35.88 kDa) (indicated by the arrow). (**e**) HT29 cell adhesion was observed under a light microscope (40×). Data are presented as the mean ± SD. Unpaired Student *t*-test was used to compare the two groups. * *p* < 0.05, ** *p* < 0.01, *** *p* < 0.001, and **** *p* < 0.0001.

**Figure 3 ijms-24-11107-f003:**
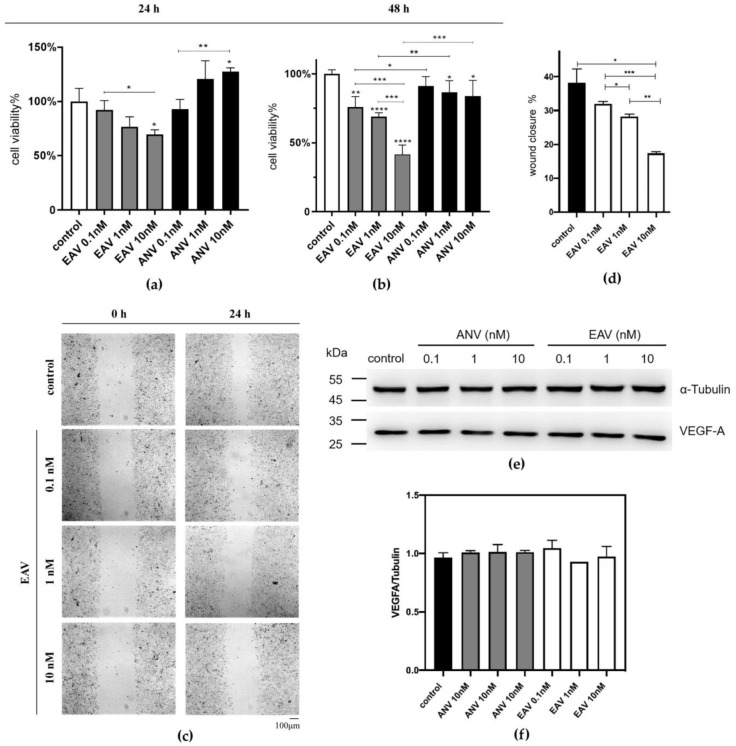
Cell viability and migration with EAV: (**a**,**b**) Effect of different protein concentrations of EAV fusion protein and ANV on B16F10 cell viability at 24 h and 48 h. The OD_570_ value was measured. (**c**) Effect of the EAV fusion protein on B16F10 cell migration. Pictures were taken with an inverted microscope at 100×. (**d**) Cell migration data statistics. The scratched area was quantified by using ImageJ 1.8.0 software. (**e**) B16F10 cells were treated with different concentrations of protein and cultured for 24 h (37 °C, 5% CO_2_). Cells were collected, and total protein and Western blotting analyses were used to detect VEGF-A expression levels in B16F10 cells. (**f**) The effect of different concentrations of ANV and EAV on the expression of VEGF-A in B16F10 cells. Normalization of gel blotting strips was performed by using ImageJ 1.8.0 software. Data are presented as the mean ± SD. *p* values derived using one-way ANOVA for multiple data sets. * *p* < 0.05, ** *p* < 0.01, *** *p* < 0.001 and **** *p* < 0.0001.

**Figure 4 ijms-24-11107-f004:**
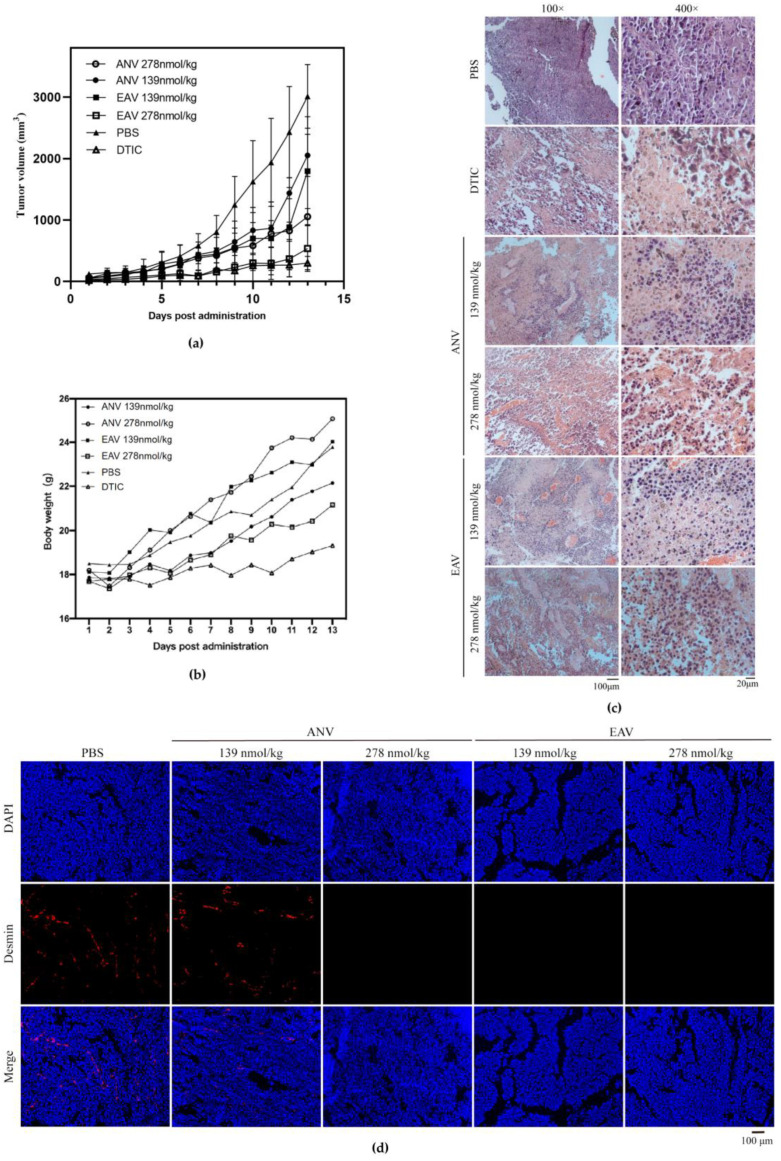
Antitumor effects of EAV in mice bearing B16F10 melanomas: (**a**) Tumor volumes were measured every day from Day 1 to Day 13. The mean ± SD was calculated from the tumor volume of 5 mice in each group (mm^3^). (**b**) The body weights of six groups of mice were measured daily from Day 1 to Day 13. (**c**) Comparative histological analysis of tumor necrosis between different groups via H&E staining. The melanoma tissue blocks were cut into 5 μm slices and dried in a 60 °C incubator for 1 h. The sections were dewaxed and rehydrated. H&E staining was performed, and the sections were observed with an inverted microscope. (**d**) Detection of tumor angiogenic inhibition by using tumor vascular fluorescence imaging. Antigen repair was performed after producing tumor sections. The sections were incubated with Desmin primary antibody overnight at 4 °C and Alexa Fluor 555-labeled secondary antibody at room temperature for 2 h, after which they were sealed with a blocker containing DAPI. Images were taken with an inverted fluorescence microscope viewed at 100× (DAPI excitation wavelength was 340 nm, and Alexa Fluor 555 excitation wavelength was 555 nm).

**Figure 5 ijms-24-11107-f005:**
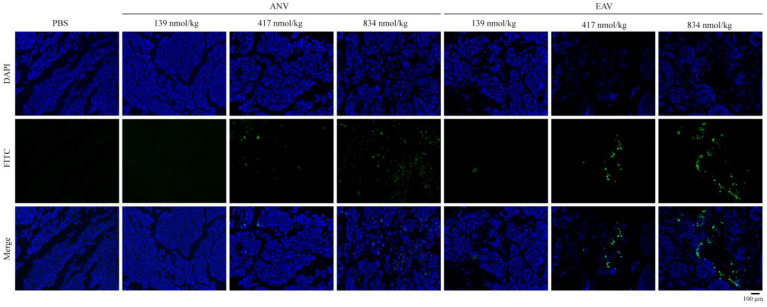
Targeting of EAV to melanoma tissue. FITC-labeled ANV and EAV were injected into mice via the tail vein; half an hour later, tumor tissue sections were produced. Magnification, 100× with an inverted light microscope.

**Table 1 ijms-24-11107-t001:** CD determination of EAV and ANV secondary structures. The table displays the secondary structure species and ratios of the EAV fusion protein and ANV.

	Fraction	Number	Ratio (%)
ANV	α-helix	231	72.41
	β-turn	8	2.51
	extended strand	6	1.88
	random coil	74	23.20
EAV	α-helix	237	59.25
	β-turn	16	4.00
	extended strand	14	3.50
	random coil	133	33.25

## Data Availability

The data presented in this study are available on request from the corresponding author.

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
