# Peer review of "Design, Synthesis and Anti-Melanoma Activity of Novel Annexin V Derivative with β3-Integrin Affinity"

_ijms, 2023, doi:10.3390/ijms241311107_

Round 1

Reviewer 1 Report

Comments to the Author

The content of this manuscript is well-designed and well-explains the purpose, method, and results of the study. In addition, results strongly supports Annexin V derivative with beta3-intergrin affinity has anti-melanoma activity.

There are some minor points which need to be considered for improving clarity of the MS.

1. Please describe the description of lane 4 in the legend of Fig.1(d).

2. Fig. 2 (b), please clearly indicate the comparison groups in your statistical analysis on the graph.

3. Fig. 2 (b) and (c) are shown as experimental results for HT29 cell, but there is no explanation for this in the manuscript (page 4, lanes 149-155). This part needs improvement.

Author Response

The content of this manuscript is well-designed and well-explains the purpose, method, and results of the study. In addition, results strongly supports Annexin V derivative with beta3-intergrin affinity has anti-melanoma activity.

There are some minor points which need to be considered for improving clarity of the MS.

  1. Please describe the description of lane 4 in the legend of Fig.1(d).

Response

Thanks for your advice. We have added a description of lane 4 in legend of Fig. 1, lane 4 is ANV.

  1. 2 (b), please clearly indicate the comparison groups in your statistical analysis on the graph.

Response

We have added a description of the comparison group in legend of Fig. 2 (page 6, lanes 182-184 of the revised manuscript).

  1. Fig. 2 (b) and (c) are shown as experimental results for HT29 cell, but there is no explanation for this in the manuscript (page 4, lanes 149-155). This part needs improvement.

Response

Thanks for your advice. We have added a description of the experimental results of Fig. 2(b) and (c) in lanes 165-171 of the revised manuscript.

Reviewer 2 Report

In this study an Echistatin-Annexin V (EAV) fusion protein possessing a high degree of αvβ3 and αIIbβ3 integrin receptor recognition and binding characteristics while retaining the specific binding ability of the natural ANV molecule for phosphatidylserine (PS) was synthesized and evaluated in a mouse xenograft model. The authors show that  EAV inhibited the viability and migration of B16F10 murine melanoma cells in a dose-dependent manner, exhibited good tumor suppressive effects in a xenograft mouse melanoma model with the inhibition of angiogenesis. EAV exhibited stronger biological effects than natural ANV molecules in inhibiting melanoma in mice. The authors suggest that the unique biological effects of EAV are based on its high β3-type integrin receptor-specific recognition and binding ability, as well as its highly selective binding to PS molecules. Based on these findings, the authors propose that EAV-mediated tumor suppression is a novel and promising antitumor strategy.

On the whole, the study is well scheduled and performed, generating convincing results. The fusion protein strategy has merit and deserves to be further evaluated.

Comments

Even though the authors show data regarding the binding of cells to different substrates (e.g. Col-1 , FN) there is very little mention of the extracellular matrix and its role in cancer and especially melanoma progression. Therefore, this needs to be further introduced and discussed in respective sections.

Minor points

line 39, platelets are not cells

line 59, clarify

Minor text language edits are necessary throughout.

Minor text language edits are necessary throughout.

Author Response

Comments

Even though the authors show data regarding the binding of cells to different substrates (e.g. Col-1 , FN) there is very little mention of the extracellular matrix and its role in cancer and especially melanoma progression. Therefore, this needs to be further introduced and discussed in respective sections.

Response

The interactions of the extracellular matrix with vascular endothelial cells and αvβ3 are reflected in the yellow highlighted areas of the revised manuscript discussion section (page 11, lanes 338-347 of the revised manuscript).

Minor points

line 39, platelets are not cells

Response

Corrected, please see line 42 on page 1 of the revised manuscript.

line 59, clarify

Response

Clarified, please see line 60-62 on page 2 of the revised manuscript.

Minor text language edits are necessary throughout.

Response

We have invited native English speaker to read this article and ensure the standardization and fluency of the English language.

Response

The data on cell binding to different extracellular matrix components (Col-1, FN) were aimed at detecting the integrin receptor binding characteristics of EAV. Through the study, we confirmed that EAV possesses good αVβ3 binding properties, but not α1β1. This is very important for understanding the molecular mechanism of EAV against melanoma.

The role of extracellular matrix in melanoma progression is a very interesting topic. We have added an introduction to the role of β3 integrin and ECM (extracellular matrix) interactions during melanoma progression in the discussion section, mainly regarding signaling pathways. It is also our future research priority to reveal the mechanism of melanoma inhibition by EAV.
